# The Establishment of an Ex Situ Collection of *Primula veris* in Bulgaria

**DOI:** 10.3390/plants11223018

**Published:** 2022-11-08

**Authors:** Elina Yankova-Tsvetkova, Maria Petrova, Irena Grigorova, Boryanka Traykova, Marina Stanilova

**Affiliations:** 1Department of Plant and Fungal Diversity and Resources, Institute of Biodiversity and Ecosystem Research, Bulgarian Academy of Sciences, Acad. G. Bonchev str., 23, 1113 Sofia, Bulgaria; 2Institute of Plant Physiology and Genetics, Bulgarian Academy of Sciences, Acad. G. Bonchev str., 21, 1113 Sofia, Bulgaria

**Keywords:** cowslip, seed germination, gibberellic acid, monochromatic lights

## Abstract

*Primula veris* is a valuable medicinal plant species with declining populations, protected in Bulgaria by the Biodiversity Act. The present study aimed to increase its extremely low seed germination rate, starting with seeds originating from two Bulgarian populations, and to set up an ex situ field collection. The stimulation effect of three factors was tested in in vivo and in vitro experiments: seeds treated with gibberellic acid (in different concentrations and exposure time), light quality (white, infrared, red, and blue or dark), and cold stratification. The combination of factors resulted in 36 treatment variants in vivo and 8 treatment variants in vitro. No germination was observed in control treatment variants. The highest germinating rate (95%) was noticed in vivo under blue monochromatic light after seed soaking into 0.2% GA_3_ for 10 h; however, the best results (55% of well-developed seedlings) were observed with a combination of blue light and 0.3% GA_3_ for 5 h. Seedlings were successfully strengthened in vermiculite in a phytotron, potted in soil and grown in a greenhouse, and then 75 plants were transferred to the field plot, where most of them bloomed at the first vegetation season. These results are intended to serve as a basis for establishing a pilot agriculture of the species.

## 1. Introduction

*Primula veris* L. (Primulaceae), known as cowslip, is a herbaceous polycarpic perennial species native to Europe and western Asia [1]. The species has been used as a medicinal plant since the Middle Ages, for treatment of gout, headache, and rheumatism [2]. Today, *P. veris* is widely used for its diverse medicinal effects: secretolytic, expectorant, anti-inflammatory, diuretic, antimicrobial, antifungal, and sedative [3,4,5,6]. Presently, cowslip is less abundant in Europe than in the past, and the sustainable supply of the source material has become more difficult [7,8]. The species is included in the European Red List of Medicinal plants under the category “Least Concern” [9]; however, according to the European Pharmacopoeia, it is still used as a source of *Primula* roots together with *P. elatior* [10]. In Poland, some of the natural populations of *P. veris* are endangered as a result of plowing, grazing, digging up for decorative purposes, and the species being under partial legal protection [11]. One of the reasons causing the loss of some populations could be related to the very low genetic variation in the natural populations [12]. Another reason is the reduced reproduction capacity of the populations, which might be partly explained by the presence of an imbalance of flower morphs, leading to a lack of compatible pollen [7,13].

In Bulgaria, *P. veris* is widespread, but most of its populations are small and fragmented, and the estimated seed viability is extremely low [14]. Seed production is known to be a function of population size, i.e., seeds originating from large populations have higher germination success, which increases with increasing seed mass [7,15]. The limited size of Bulgarian populations of *P. veris* and the low seed viability create a long-term risk for its extinction [14]. The species is protected by the Biodiversity Act, Annex 4 (2002) [16] and is under a special regime of use based on the Medicinal Plants Act (2000) [17]. It is included in the list of medicinal plant species under the special regime of protection and use (Ordinance no. RD-71/2007), i.e., flowers and roots of cowslip are harvested according to the annual quotas of the Ministry of the Environment and Waters. A good alternative would be the introduction of *P. veris* in agriculture, which requires research aimed at accelerating plant reproduction.

Seed germination in *Primula* spp. has been reported to be difficult [18,19,20], and propagation by cuttings cannot be realized due to the morphology of the plants, i.e., dense rosettes of leaves close to the ground [21]. Propagation from seeds is usually cheap and effective, but in *P. veris* it is impeded by a number of factors controlling seed germination. The main of these factors is the seed dormancy, the biological purpose of which is to avoid seed germination during years with unfavorable environmental conditions. Dormancy may occur at the level of the embryo, seed coat, or seed covering [22].

Seed dormancy is controlled by the balance between two plant hormones: abscisic acid (ABA) and gibberellic acid (GA_3_) [23]. High levels of endogenous ABA in mature seeds cause their dormant state, while increases in GA_3_ levels leads to dormancy breaking [24]. The ABA:GA_3_ ratio is influenced by abiotic factors such as natural changes in temperature, moisture, or light. In laboratory conditions, the balance of the two hormones can be controlled by either treatment with exogenous GA_3_ or with cold stratification, which induces the transcription of GA_3_ biosynthesis genes, stimulating spikes in endogenous GA_3_ [25]. Seed dormancy of *P. veris* seeds was reported to be overcome in vitro owing to the application of GA_3_ [21]. In addition to seed dormancy, some factors affecting seed germination are also the age and the nutrition of the mother plants during seed maturation [26], the population size [7], the time of seed collection, and the duration of seed storage [27].

Small seeds, such as those of *P. veris*, need light to germinate because their nutrient reserves are limited and their seedlings must reach the soil surface and begin to photosynthesize before depleting seed nutrients [28]. Many authors suggest that seed mass and light requirements coevolved to ensure germination [29]. According to the light requirements of seeds for germination, plants are classified into three groups: positive, negative, and neutral photoblastic [30].

In order to ensure the mass propagation of plants from *Primula* species needed for agriculture establishment, in vitro experiments on seed germination and plantlets regeneration were carried out with *P. obconica* [18,31,32], *P. vulgaris* [33], *P. malacoides* [32], *P. cuneifolia, P. scotica, P. veris, P. latifolia, P. heterochroma* [34,35,36,37,38], and several *Primula* hybrids [39]. Shoot multiplication and in vitro rooting were obtained in some cowslips on media containing different cytokinins [34,37,40,41,42]. Recently, a protocol for the in vitro micropropagation of *P. veris* subsp. *veris* was established [21].

The aim of the present study was to increase the extremely low seed germination of *P. veris*, comparing different approaches in parallel and starting with seeds originating from Bulgarian populations, and to set up an ex situ field collection of the species. These results are intended to serve as a basis for establishing a pilot agriculture of the species.

## 2. Materials and Methods

### 2.1. Plant Material

*P. veris* plants gathered in April 2019 from two Bulgarian native populations: on Golo Bardo Mt over the town of Pernik (42.563822° N, 23.073438° E; 910 m altitude) and near the village of Ilindentsi in Pirin Mt (41.680444° N, 23.297083° E; 970–1100 m altitude), were used for the establishment of an ex situ collection on the experimental field plot of the Institute of Biodiversity and Ecosystem Research at the Bulgarian Academy of Science (IBER-BAS). Seeds from these plants and from the natural populations were collected in June–July 2020 and used for in vitro and in vivo experiments on germination immediately or after 2- or 6-month storage at room temperature or stratification at 4 °C. The obtained plants from these seeds were used to expand the ex situ collection. All trials were conducted at the Laboratory of Plant Anatomy and Embryology, the Biotechnological Laboratory of Medicinal Plants, and the experimental field plot of IBER-BAS (N 42.67115; E 023.36458, 506 m altitude).

### 2.2. Experimental Design and Statistical Analysis

Seeds were germinated in vitro and in vivo in parallel. Seeds designated for in vitro germination were first surface sterilized by standard procedure (consecutive soaking into 70% ethanol for 1 min, and commercial bleach, Cl < 0.5%, for 6 min), then rinsed three times with sterile distilled water for 5, 10, and 15 min. Seeds were put on basal MS medium [43] free of plant growth regulators (PGRs) or MS supplemented with 0.5 mg/L kinetin (Kin) and 1.0 mg/L gibberellic acid (GA_3_) (medium G1K_5_) or 5.0 mg/L GA_3_ (medium G5K_5_) (all PGRs from Duchefa, NL). Before incubating seeds on basal MS medium, they were pretreated with 0.1% GA_3_ for 20 h or with 0.3% GA_3_ for 20 h or for 5 h; in the control treatment, variant seeds were soaked in water for 20 h. In addition, 2-month cold stratification at 4 °C was applied before incubation of seeds on MS control medium and on medium G5K_5_. All media contained 30 g/L sucrose and were adjusted to pH 5.75, solidified with 6.5 g/L plant agar (Duchefa, NL), autoclaved at 121 °C, 1 atm for 20 min, and poured into containers with a grid (Duchefa, NL). Cultivation conditions in the culture room were: 23 ± 2 °C, 16/8 h light/dark period. Each treatment variant consisted of 2 repetitions, 50 seeds per repetition. Germination rate in all 8 treatment variants was assessed in percentage first at the end of the 10th week, then 5 months after seed incubation. 

In vitro sub-cultivation was performed on MS medium supplemented with 1 mg/L BAP + 0.2 mg/L 2,4-D (medium B1D_2_) using 20 in vitro seedlings with removed roots as explants. Propagation coefficient was evaluated as the number of new shoots obtained per explant. For in vitro rooting of the shoots, a half-strength MS medium containing 20 g/L sucrose and supplemented with 0.5 mg/L IBA + 1 g/L activated charcoal (medium MS_root_) was used. Cultivation conditions were the same as for in vitro seed germination.

The influence of light quality and GA_3_ on seed germination was tested in vivo as a two-step treatment, using seeds taken from the ex situ collection. Before exposure to different lights, seeds were pretreated with GA_3_. Solution of GA_3_ was applied in four concentrations (0% used as a control, 0.1%, 0.2%, and 0.3%) and three time exposures (5 h, 10 h, and 20 h) in a total of 6 joint levels (0% 20 h, 0.1% 5 h, 0.1% 20 h, 0.2% 10 h, 0.3% 5 h, and 0.3% 20 h). Six levels of illumination were tested: white light from fluorescent tube (F) used as a control, white light-emitting diodes (W), infrared (I), red (R), blue (B), and darkness (D). Combinations tested of the factors resulted in 36 treatment variants in vivo, 20 seeds per treatment variant (Table 1). Seeds were placed in petri dishes on wet filter paper, 20 seeds per petri dish. The temperature was 23 ± 2 °C, and the photoperiod 12/12 h light/dark, except for the treatment variants in darkness. The germination rate was assessed in each treatment variant at the end of the 8^th^ week. In addition, germination of seeds taken from the natural populations was tested in vivo in a growth camera at 11 ± 1 °C, and fluorescent light with the same photoperiod, after 6-month stratification at 4 °C.

Statistical analyses were performed with Excel’s two-factor ANOVA without replication. The effects of the light quality and the pretreatment with GA_3_ on the seed germination were assessed by grouping the treatment variants with equal levels of these factors, thus evaluating the relative importance of the light and the gibberellic acid. Since a lot of seeds stopped their development at the embryonic root stage, the same analysis was performed taking into account only the seedlings with cotyledons or/and first pair of true leaves. In addition, differences between the levels of one and the same factor were assessed with Excel *t*-test paired analysis. 

### 2.3. Establishment of an Ex Situ Collection

To establish the ex situ collection of *P. veris* on the experimental field plot of IBER-BAS in Sofia (506 m altitude), ten plants were used gathered from two natural Bulgarian populations (on Golo Bardo Mt, over the town of Pernik, and near the village of Ilindentsi in Pirin Mt), five individuals from each one. They were planted in two flower beds, at a distance of 40 cm between them, and cultivated under controlled conditions: watering, weeding, etc.

Viable in vivo obtained seedlings were transferred to containers with perlite or vermiculite for 3 months. Surviving plants were potted in soil mixture (light soil, sand, and coconut fibers in proportion 2:1:1) for further adaptation in a room phytotron with a window, then in the greenhouse. Cultivation conditions in the phytotron varied around the clock: temperature between 18 and 26 °C, relative air humidity from 30% to 60%, mixed daylight and artificial white LED 16/8 h light/dark. Plants were planted on the ex situ collection twice: in late September 2021 and in early April 2022.

## 3. Results

### 3.1. In Vitro Cultivation 

#### 3.1.1. In Vitro Germination

Our first trials to germinate seeds gathered from two Bulgarian natural populations of *P. veris*, immediately after their collection in July 2019 and after storage of 6 months at room temperature, were unsuccessful, as only one seed germinated from 400 incubated, regardless of the medium composition. Seeds also did not germinate in a growth camera at 11 ± 1 °C, after 6-month stratification at 4 °C. Therefore, the next experiments were carried out with seeds collected from the plants growing in the ex situ collection originating from the Golo Bardo population, which was chosen as a model population.

Stratification was of crucial importance for the successful germination under in vitro conditions. The most successful factor combination was a 2-month seed stratification at 4 °C followed by seed pretreatment with 0.1% GA_3_ for 20 h prior to seed incubation on medium G5K_5_—50% germinated seeds. With the same factor combination, on the basal MS medium free of PGRs, the germination rate was 20% (Figure 1A). 

Among treatment variants without cold stratification, seed germination was relatively high when GA_3_ was added to the medium: 36% on medium G1K_5_, while in the treatment variants with seed pretreatment with GA_3_ solution, the percentages of the germinated seeds varied between 4% and 22% according to the concentration of the solution and the time of exposure. No seeds germinated in the control treatment variant. It is worth noting that a total of 29% of all germinated seeds necrotized and that seedlings with a pair of true leaves developed only in two treatment variants: 12% on medium G1K_5_ and 8% on MS medium when seeds were pretreated with 0.3% GA_3_ for 20 h.

#### 3.1.2. Sub-Cultivation and In Vitro Rooting

In vitro seedlings with removed roots were sub-cultured on medium B1D_2_. The survival rate after 2 months of cultivation was 90%, and almost all explants formed 2 to 5 adventitious shoots (Figure 1B,C). The propagation coefficient was 3.05 shoots per explant (Figure 1D). Leaves of few shoots turned white. Shoots were separated from one another and put on medium M_root_ for rooting. After 6 weeks, 34.4% of the shoots necrotized, and only 3.1% of the surviving shoots formed rudiments of roots, but they were not suitable for ex vitro adaptation (Figure 1E). 

### 3.2. In Vivo Germination and Seedlings Growth

The first attempts to germinate seeds gathered from the natural populations of *P. veris* were unsuccessful. As no seeds germinated from a total of 400 used, the next trials were undertaken with seeds collected from the plants growing in the ex situ collection, similar to the in vitro experiments.

The cold stratification had no effect on seed germination: no germination was observed after the storage of seeds at 4 °C over 6 months.

#### 3.2.1. Effect of GA_3_ and Monochromatic Lights on Germination Efficiency

The variation in the germination rate following the treatment variants tested is presented in two diagrams corresponding to the factors GA_3_ pretreatment, with jointly presented levels of concentration and exposure time of GA_3_ (Figure 2A), and light (Figure 2B).

The pretreatment with GA_3_ before seed exposure to different lights was crucial for the successful seed germination. When seeds were soaked in water instead of GA_3_ solutions, no seeds germinated except in the dark (treatment variant CD), where only 15% of the seeds germinated and all stopped developing at the embryonic root stage. All combinations of GA_3_ concentration and exposure time stimulated seed germination; however, relatively high concentrations and shorter exposure times appeared to be most favorable concerning seedling quality.

The effect of the monochromatic lights was positive or negative depending on their wavelengths. The highest germination rates were noticed when seeds were incubated under blue or red light. In the best treatment variants with blue light, up to 95% (i.e., 19 from 20 seeds) and 65% (i.e., 13 from 20 seeds) of the seeds germinated after soaking into 0.2% GA_3_ for 10 h and into 0.3% GA_3_ for 5 h, respectively. In the case of red light, up to 60% of the seeds germinated in treatment variants 0.1R5, 0.1R20, and 0.2R10. However, red light caused the etiolation and death of many seedlings, while blue light stimulated the development of large cotyledons and leaves (Figure 3). Survival rate depended on the formation of cotyledons and the first pair of leaves, which was mostly stimulated by the blue light. 

The results for seed germination and seedling development in each treatment variant are presented in Table 2. The statistical analyses showed significant differences among treatment variants in terms of germination rate (from 0% to 95%) between the two factors tested (*p* < 0.001 concerning GA_3_ pretreatment and *p* < 0.01 concerning lights) (Table 2(A)). In 13 treatment variants, half or more of the seeds germinated. However, in all treatment variants, a number of seeds stopped their development at root stage and were eliminated. Differences between treatment variants remained significant regarding the percentage of the seedlings with cotyledons or with cotyledons and first pairs of true leaves (*p* < 0.001 for GA_3_ pretreatment and *p* < 0.05 for light), ranging between 0% and 55% of seeds developing into seedlings (Table 2(B)).

The best seedling development was observed when seeds were pretreated with higher concentration of the gibberellic acid for a short time (0.3% GA_3_ for 5 h), when most seeds formed plants; in all other groups, seeds dying at stage root emergence predominated (Figure 2A). Concerning light quality, the best proportion between seeds developing into plants and seeds dying after root formation was observed in the cases of white and blue light, but the percentage of the well-developed seedlings was twice as high under the blue monochromatic light (Figure 2B). More detailed results are presented in Table 3 and Table 4, comparing each pair of factors’ levels using Excel *t*-test paired analysis. Finally, treatment variant 0.3B5 (the combination of 0.3% GA_3_ for 5 h, and blue light) was the best one, ensuring the developing of 55% of the incubated seeds into well-shaped seedlings with large green leaves. Among the other treatment variants tested, relatively good results were obtained in 0.2B10 (0.2% GA_3_ for 10 h and blue light) and 0.3W5 (0.3% GA_3_ for 5 h and white light), both with 40% seeds developing into well-shaped seedlings.

#### 3.2.2. Seedlings’ Growth and Survival

Perlite was found to be an unsuitable substrate for seedlings’ adaptation to the phytotron conditions, as all 20 seedlings transferred to a container with perlite died. The next 79 seedlings were transferred to vermiculite, and all of them grew and strengthened after three months (Figure 4A). Then they were potted in soil mixture, and after another three months in the phytotron, 94.9% of them survived and formed new leaves (Figure 4B). Plants were transferred to the greenhouse for acclimation before planting in the ex situ collection.

### 3.3. Establishment of an Ex Situ Collection

All plants brought from the natural populations in 2019 were successfully acclimated to the conditions of the field plot of the Institute of Biodiversity and Ecosystem Research (IBER-BAS) and bloomed the next years. They produced numerous seeds that were bigger in size than those collected from the natural populations (Figure 5).

Plants grown from seeds and strengthened in the greenhouse were also transferred to the field plot to expand the ex situ collection of *P. veris*. The success of their acclimation differed according to the season of planting: 68.8% of 32 plants transferred in autumn 2021 survived the wintering, while the survival rate of those planted in spring 2022 (43 plants) increased up to 93%. Most of the plants bloomed in April–May 2022 in spite of their relatively small size (Figure 6).

## 4. Discussion

The lack of germination of the seeds taken from *P. veris* plants growing in the two Bulgarian natural populations is consistent with the extremely low embryo viability previously reported for seeds of the same origin: 4% and 2% for the ‘Golo Bardo’ and ‘Ilindentsi’ populations, respectively [14]. Authors stated that half to ¾ of all seeds were empty. The high number of empty seeds is probably related to insufficient pollen at the high altitudes at which *Primula* populations are located. It is proven that inadequate quantities or quality of pollen can reduce plant reproductive success (seed quantity or quality), known as pollen limitation [44]. In perennial self-incompatible plant species (such as *Primula*), inadequate pollen supply is a major cause of low fertility [45]. A correlation between population size and pollen limitation was also found. According to Ward and Johnson [46], the reduced seed production and the fragmentation of small populations of *Brunsvigia radulosa* are attributable to pollen limitation. Agren [47], studying the island populations of heterostylous *Lythrum salicaria*, found that the degree of pollen limitation increased with the decreasing of the population size. The same is the case with *Primula veris*, which is a heterostylous species with small and fragmented populations.

Yankova-Tsvetkova et al. [14] established that the percentages of viable seeds of the plants transferred to the ex situ collection increased significantly in only one year: up to 68% and 76%, respectively. Under the controlled conditions of the field plot (watering, weeding, more pollinators), plants produced larger seeds, and only 2% of the seeds were empty [14]. This explains the potential to stimulate the germination of the seeds produced ex situ. Moreover, increasing seed mass was reported to favor the germination ability [10]; seeds of field-cultivated plants had more chance to germinate.

Similar results showing no germination potential of seeds collected from wild populations of *P. veris* subsp. *veris* were also noticed by Grigoriadou et al. [21], who used seeds stored in a seed bank for 8, 14, and 17 years. These authors suggested that long seed maintenance was the most likely reason for this; however, the difficult germination was highlighted as a characteristic for *Primula* seeds in other investigations as well [18,19,20]. In natural populations of *P. veris*, which is a heterostylous perennial species, seed set and seed size have been shown to depend on many factors such as population size, presence and reciprocal ratio of plants with different flower morphs, pollinator availability, presence of pathogens and herbivores, and other environmental or genetic factors [7,48,49,50]. Jedrzejczyk et al. [51] found out that not only the origin but also the year of seed collection significantly affected seed quality in cowslip plants derived from seeds or regenerated in vitro. In this relationship, the small and fragmented Bulgarian populations do not favor seed germination. Seed quality of many cowslips and primroses is affected by bacterial and fungal infections, resulting in losses of seeds and breading material [41].

Another factor impeding germination is the strict seed dormancy of *P. veris* found by Milberg [52], who pointed to cold stratification as a requirement for overcoming it. According to his research, seed capacity to germinate varied over the year, exhibiting a dormancy cycle typical for many species from temperate areas, and the best germination was recorded at 16 °C under a photoperiod of 12/12 h dark/light. Brys and Jacquemyn [53] noticed that mature seeds were released from plants in summer and most of them overwintered and germinated in April–May. In our case, despite the large increase in seed viability under controlled cultivation conditions, from 2% to 76% [14], seeds failed to germinate without stimulation due to their dormancy. Similarly, Milberg [52] reported that over 80% of the ungerminated seeds were viable according to the tetrazolium test, and in many seeds, the secondary dormancy was overcome the next winter.

In the present study, seed dormancy was successfully broken under laboratory conditions in vitro and in vivo using gibberellic acid as a stimulant. The effect of gibberellins on cowslip seed germination was reported by other authors as well [7,20,21]. Morozowska [20] stated that the effectiveness of GA_3_ treatment was influenced by the germinating temperature and increased most at 10 °C, especially after a 7-day prechilling. Our results showed that the addition of GA_3_ in the medium was more effective then seed pretreatment with this growth regulator, but in both cases, cold stratification was required. The same conclusion concerning the effect of GA_3_ was reached by Grigoriadou et al. [21], irrespective of the photoperiod regime. The presence of kinetin in the medium seemed to be suitable for seedlings’ development. Morozowska and Wesołowska [36] obtained the largest number of *P. veris* subsp. *veris* seedlings on MS medium supplemented with GA_3_ and kinetin, which was also observed in our study. Seeds of *P. boveana* also did not germinate on media free of growth regulators, while the presence of GA_3_ led to germination rates ranging from 10% to 77% [54]. Some authors [21] also added GA_3_ in the medium for shoot multiplication together with BAP and IBA and chose it as the best one; however, the propagation rate was similar to that obtained in our experiments on medium containing Kin and IBA. The same authors successfully rooted in vitro regenerated shoots on media with different concentrations of IBA, but in spite of the increased root number, the root length was optimal on the control medium, as IBA had an inhibitory effect. Apparently, the concentration of IBA in our rooting medium was higher than the optimal one for this species, which impeded the elongation of the root rudiments formed. Further experiments are needed to precisely determine the optimal concentration of this plant growth regulator in order to improve the in vitro rooting and to achieve the ex vitro adaptation of the plantlets. According to some authors who carried out a comparative morphological analysis of in vitro regenerated and seed-derived cowslip plants, the latest were characterized by more intensive development rate and higher total seed production [51].

The influence of the light quality on seed germination ability was also determined in the present study, under in vivo conditions. Fluorescent white light was chosen as the control, as it had been found that small seeds generally germinate better in light than in the dark because they remain on the soil surface after being released from the plants [29]. Baskin and Baskin [30], in their classification of plants in terms of their responses to light concerning germination, defined *P. veris* as a positive photoblastic plant that requires light to germinate. In the present in vivo experiment, many seeds germinated in the dark, but they stopped their development at the stage of root emergence. The development of seedlings was possible only when the two stimulating factors light and GA_3_ were applied together. The combination of blue monochromatic light and pretreatment with 0.3% GA_3_ for 5 h was found to be the most successful among all 36 combinations tested. The seed dormancy of *Alkanna tinctoria* was also successfully overcome by successively stimulating seeds with seed pretreatment with GA_3_ solutions and monochromatic red or blue light [55]. The effect of monochromatic lights was tested on different species and was found to be species dependent and to either stimulate or inhibit seed germination [56,57,58]. In *Halophila ovalis*, the red light not only enhanced seed germination but also favored seedling survival [58], which was in contrast with our results, as red light caused the etiolation and death of many seedlings of *P. veris*.

The acclimation of all propagated plants in the present study was very successful not only in terms of plant survival rate but also in terms of their maturation, as many plants flowered in their first growing season, regardless of the time (autumn or spring) of their transfer to the ex situ collection. This result was much better than the data reported by Brys and Jacquemyn [53] claiming that *P. veris* seedlings need at least three years to start flowering.

## 5. Conclusions

The native, extremely low seed germination of *Primula veris* can be overcome. Under the controlled conditions of an experimental field plot, the ability of seeds to germinate increases. It is additionally improved by seeds’ pretreatment with gibberellic acid and stimulation with monochromatic light. A protocol is established for the cultivation of the species, including the conditions for seed germination, seedling strengthening, and the gradual acclimatization of the plants outdoors. Transferring the plants to the experimental field plot is more appropriate in the spring. The results of the present study are intended to serve as a basis for establishing a pilot agriculture of the species.

## Figures and Tables

**Figure 1 plants-11-03018-f001:**
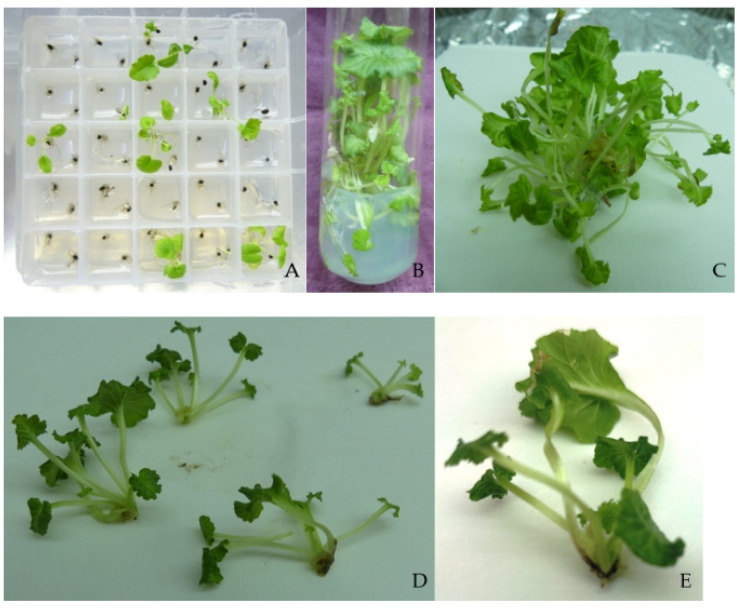
In vitro cultivation of *P. veris*: (**A**) Seed germination on medium G1K_5_; (**B**,**C**) Shoot-clump after sub-cultivation on medium B1D_2_.; (**D**) Separated shoots from the shoot-clump; (**E**) Shoot with rudiments of roots.

**Figure 2 plants-11-03018-f002:**
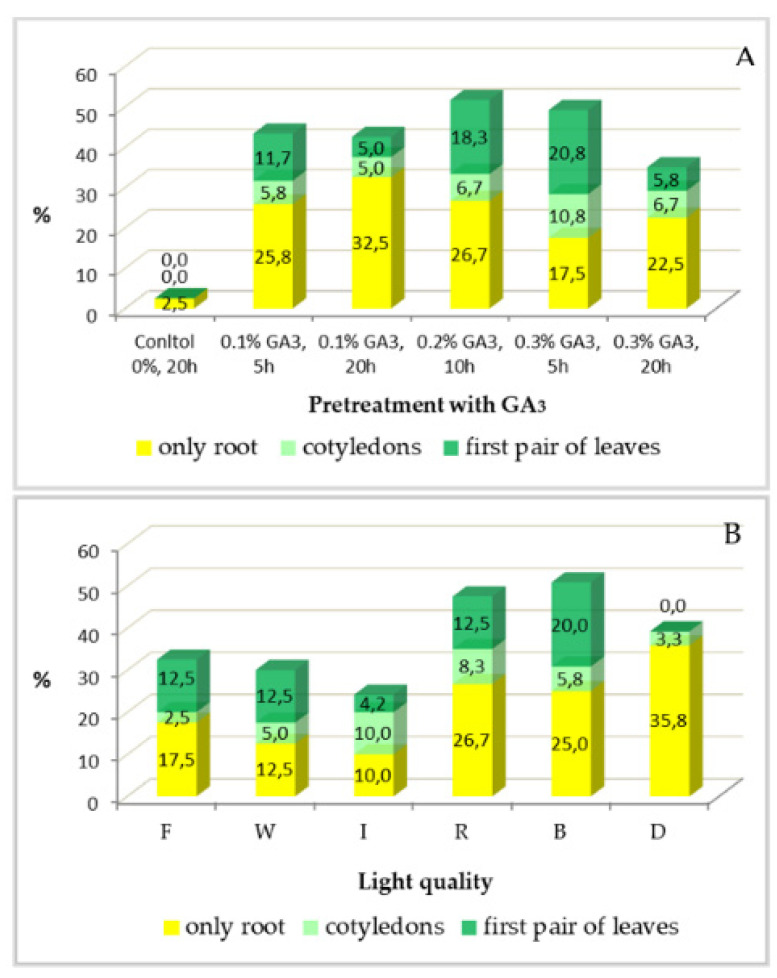
Percentages of germinated seeds and seedlings of *P. veris* at early stages in the groups of treatment variants with equal levels of the factors: (**A**) GA_3_-pretreatment; (**B**) light quality (F—fluorescent white; W—white LED; I—infrared; R—red; B—blue; D—darkness).

**Figure 3 plants-11-03018-f003:**
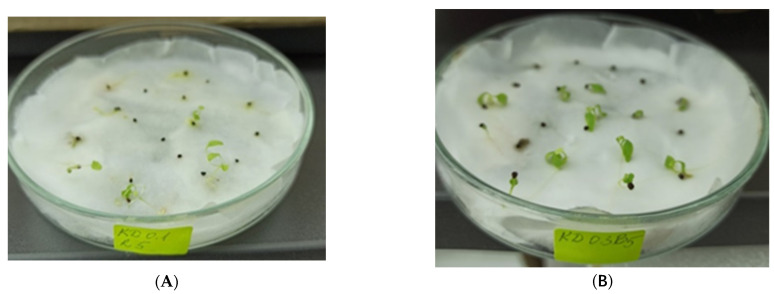
Seedlings of *P. veris* germinating in vivo under: (**A**) red light; (**B**) blue light.

**Figure 4 plants-11-03018-f004:**
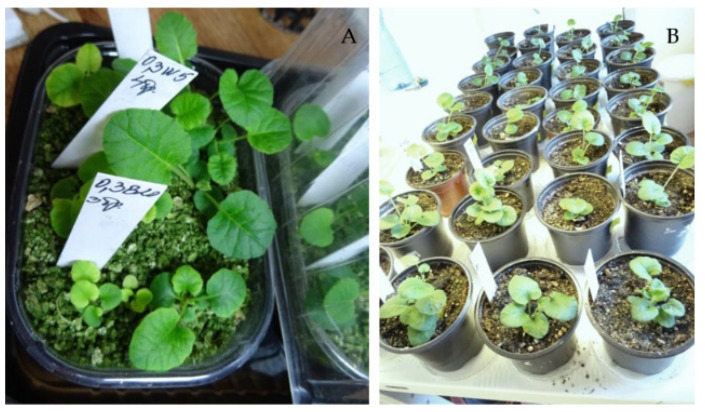
Plant adaptation of *P. veris* and growth in the phytotron: (**A**) in vermiculite; (**B**) in soil mixture.

**Figure 5 plants-11-03018-f005:**
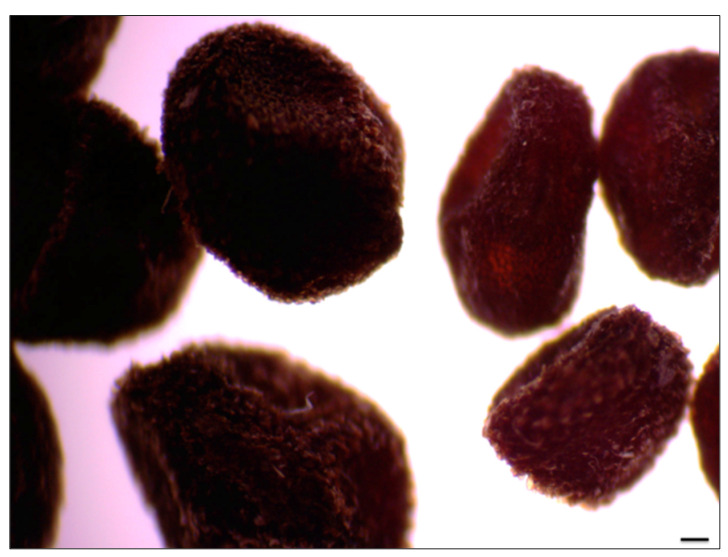
Seeds of *P. veris* gathered from the ex situ collection (the bigger ones, left) and from the corresponding natural population (the smaller ones, right). Scale bar = 100 µm.

**Figure 6 plants-11-03018-f006:**
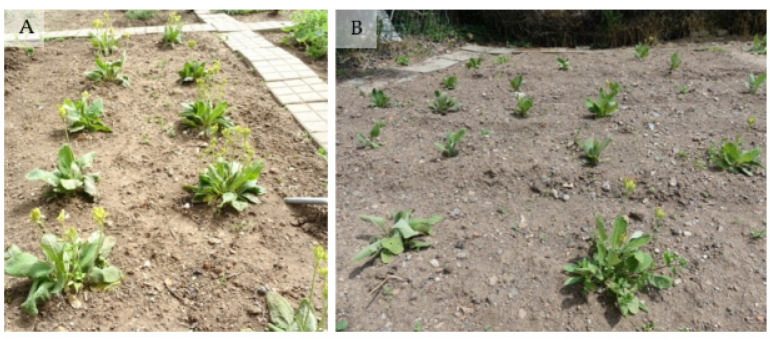
Ex situ collection of *P. veris*: (**A**) Plants brought from the natural populations; (**B**) Plants grown from seeds and transferred to the experimental field plot.

**Table 1 plants-11-03018-t001:** Treatment variants tested for in vivo seed germination of *P. veris*: codes show the factors tested (light and pretreatment with solutions of GA_3_) and their levels.

Treatment Variants ^1^	Control 0% 20 h	GA_3_ 0.1% 5 h	GA_3_ 0.1% 20 h	GA_3_ 0.2% 10 h	GA_3_ 0.3% 5 h	GA_3_ 0.3% 20 h
F	CF	0.1F5	0.1F20	0.2F10	0.3F5	0.3F20
W	CW	0.1W5	0.1W20	0.2W10	0.3W5	0.3W20
I	CI	0.1I5	0.1I20	0.2I10	0.3I5	0.3I20
R	CR	0.1R5	0.1R20	0.2R10	0.3R5	0.3R20
B	CB	0.1B5	0.1B20	0.2B10	0.3B5	0.3B20
D	CD	0.1D5	0.1D20	0.2D10	0.3D5	0.3D20

^1^ Light level: F—fluorescent white; W—white LED; I—infrared; R—red; B—blue; D—darkness. Gibberellic acid: concentration and exposure time of GA_3_ are jointly presented.

**Table 2 plants-11-03018-t002:** Percentages of germinated seeds and seedlings with cotyledons or first pairs of true leaves in each variant. Effect of light quality and GA_3_ pretreatment on: seed germination (**A**), formation of seedlings with first pair of true leaves or with cotyledons (**B**), estimated with ANOVA two-factor without replication.

Treatment Variants ^1^	Germinated Seeds (%)	Seedlings with Cotyledons or First Pair of Leaves (%)	Variants ^1^	Germinated Seeds (%)	Seedlings with Cotyledons or First Pair of Leaves (%)
CF	0	0	0.2F10	40	25
CW	0	0	0.2W10	50	30
CI	0	0	0.2I10	25	20
CR	0	0	0.2R10	60	20
CB	0	0	0.2B10	95	40
CD	15	0	0.2D10	40	15
0.1F5	50	35	0.3F5	40	10
0.1W5	35	25	0.3W5	45	40
0.1I5	5	5	0.3I5	45	35
0.1R5	60	15	0.3R5	50	50
0.1B5	55	30	0.3B5	65	55
0.1D5	55	0	0.3D5	50	0
0.1F20	35	5	0.3F20	30	15
0.1W20	25	5	0.3W20	25	5
0.1I20	50	15	0.3I20	20	15
0.1R20	60	20	0.3R20	55	20
0.1B20	40	10	0.3B20	50	20
0.1D20	45	5	0.3D20	30	0
(A)					
Source of Variation	Df		MS	F	*p*-value
Rows (Lights)	5		647.3611	3.92274	0.009204
Columns(GA_3_)	5		1952.361	11.8305	6.1 × 10^−6^
Error	25		165.0278		
(B)					
Source of Variation	Df		MS	F	*p*-value
Rows (Lights)	5		344.4444	3.090728	0.026262
Columns(GA_3_)	5		759.4444	6.814556	0.000391
Error	25		111.4444		

^1^ Codes of treatment variants are according Table 1 (Material and methods).

**Table 3 plants-11-03018-t003:** Differences between the groups of treatment variants with the same light and different levels of GA_3_ pretreatment for seedlings with cotyledons/first true leaves of *P. veris* (Excel *t*-test paired analysis).

Treatment Variants ^1^	Control-0%	0.1%GA_3_, 5 h	0.1%GA_3_, 20 h	0.2%GA_3_, 10 h	0.3%GA_3_, 5 h	0.3%GA_3_, 20 h
Control-0%		0.036161 *	0.0117248 **	0.0010150 ***	0.0169895 **	0.0136162 **
0.1%GA_3_, 5 h			0.3566104	0.1366175	0.197023	0.4260710
0.1%GA_3_, 20 h				0.0266925 *	0.0375396 *	0.3632174
0.2%GA_3_, 10 h					0.4076765	0.0221184 *
0.3%GA_3_, 5 h						0.0454943 *
0.3%GA_3_, 20 h						

^1^ Significant differences between treatment variants: * *p* < 0.05; ** *p* < 0.01; *** *p* < 0.001.

**Table 4 plants-11-03018-t004:** Differences between the groups of treatment variants with the same GA_3_ pretreatment and different levels of light for seedlings with cotyledons/first true leaves of *P. veris* (Excel *t*-test paired analysis).

Treatment Variants ^1^	F (Control)	W	I	R	B	D
F (Control)		0.6951923	0.9221064	0.5139400	0.2005223	0.0777520 ^#^
W			0.5658929	0.5160497	0.0198680 *	0.0814588
I				0.0623524 ^#^	0.0907096 ^#^	0.1005517
R					0.3143726	0.0583293 ^#^
B						0.0380039 *
D						

^1^ Significant differences between treatment variants: * *p* < 0.05; ^#^—close to a significant difference.

## Data Availability

Not applicable.

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
