# Peer review of "The Establishment of an Ex Situ Collection of Primula veris in Bulgaria"

_plants, 2022, doi:10.3390/plants11223018_

Round 1
Reviewer 1 Report
Dear authors,
The article contains useful and original information concerning germination of Primoula's seeds in various conditions and especially in relation to the effect of light and GA3 . However, some parts need to be improved, especially in the way the results are presented.
I suggest to be submitted again by the authors after major revision.
Possible suggestions:
Line 117 – Pls use the same unit for time period (days or month)
Line 124- in vitro - italics
Line 167 – in vitro - italics
3.1.1. In Vitro Germination – Results for in vitro germination are given only with %. Pls add a table or a diagram with the relative statistical analysis for each in vitro treatment
Figure 2 – Pls add indication of statistical analysis into the figure
Tables 2-5 refer to the statistical differences between the treatments giving details not necessary for the reader and confusing. I suggest to combine the Appendix A with the results of the statistical analysis indicating clearly with a symbol (eg a letter) in which treatments we have statistical differences. May be a different statistical analysis program could be used.
Discussion. pollinator availability probably is the possible explanation of “empty seeds”. In high altitudes there are not many insects – pollinators and may be this is an explanation. Pls discuss further adding special literature
Author Response
Rewiew 1
Dear authors,
The article contains useful and original information concerning germination of Primoula's seeds in various conditions and especially in relation to the effect of light and GA3 . However, some parts need to be improved, especially in the way the results are presented.
I suggest to be submitted again by the authors after major revision.
Dear Reviewer 1, Thank you for your comments and suggestions that helped improve the manuscript.
Possible suggestions:
Line 117 – Pls use the same unit for time period (days or month)
Answer: We didn’t use days; we used weeks or months, to be most accurate (8 weeks are less than 2 months, 10 weeks are more than 2 months, while 6 months is too long period to be written in weeks).
Line 124- in vitro - italics
Answer: Thanks, it is done.
Line 167 – in vitro - italics
Answer: Thanks, it is done.
3.1.1. In Vitro Germination – Results for in vitro germination are given only with %. Pls add a table or a diagram with the relative statistical analysis for each in vitro treatment
Answer: The in vitro results are deliberately presented briefly. Our aim was to increase the extremely low seed germination in Primula veris, to obtain well developed plants and to acclimatize them outdoors; to achieve this aim we tried different approaches. In this paper we put the emphasis on in vivo experiments as we achieved complete plant propagation and established a protocol, which could be further applied. In contrast, the results from in vitro experiments were less successful and need additional improvement (despite the relatively high germination in one of the variants, 50%, the great part of the germinated seeds necrotized, the propagation rate of the sub-cultivation of seedlings was relatively low, and in vitro rooting was poor). Until now we have no established a protocol for in vitro propagation of the species, but to our opinion the obtained results could be optimized. So, detailed results would be published separately after additional research, when we complete the in vitro protocol including acclimatization of the plants.
Figure 2 – Pls add indication of statistical analysis into the figure
Answer: Please, note that statistical analysis is presented in Table 2A,B and Tables 3 & 4.
Tables 2-5 refer to the statistical differences between the treatments giving details not necessary for the reader and confusing. I suggest to combine the Appendix A with the results of the statistical analysis indicating clearly with a symbol (eg a letter) in which treatments we have statistical differences. May be a different statistical analysis program could be used.
Answer: We summarized data from Appendix A in the new Table 2, using the codes of all variants (as given in Table 1 in Material and methods). In this table, seedlings with cotyledons and seedlings with first true pair of leaves are presented together in one column, as all of them developed into plants (some seeds took more time to germinate and to form true leaves). We removed not necessary data from ex tables 2 and 3, and statistical data are now presented in (A) and (B) of the new Table 2, concerning seed germination and seedling development, respectively. Ex tables 4 and 5 are now Tables 3 and 4. To avoid confusion we optimized the way of data presentation (asterisks mark corresponding numbers) and formulated better the titles of the tables. These tables complete the information concerning the effect of the different levels of the two factors tested.
Discussion. pollinator availability probably is the possible explanation of “empty seeds”. In high altitudes there are not many insects – pollinators and may be this is an explanation. Pls discuss further adding special literature
Answer: Thank you for the remark, we added in Discussion a part concerning the relation between pollinator deficiency and empty seeds presence.
Reviewer 2 Report
The staff has carried out a complete work that associates the germination protocol with the cultivation protocol, which is necessary when treating species at risk of extinction.
Author Response
Rewiew 2
The staff has carried out a complete work that associates the germination protocol with the cultivation protocol, which is necessary when treating species at risk of extinction.
Answer: Thank you for your appreciation of our work.
Reviewer 3 Report
I did enjoy reading this article. The authors bring novel insights in conservation eco-physiology of Primula veris. However, some technical corrections are needed. English improvement is also necessary.
The term ex situ collection is used inappropriate, with idea that only outdoor beds are referring to ex situ. In fact, all done and mentioned in manuscript is a mode of ex situ conserving. It needs to be adjusted.
Introduction is giving very few information of the threats of this species globally and in Bulgaria. It should be added. On the other hand it gives too much space on medicinal properties of target species.
The results presented in rather agricultural way and the aim and focus should be on conservation of the species Primula veris. But it is a matter of the style by the authors.
Conclusion cannot use future tense. The assumption on natural populations of target species recovery in this study cannot be the conclusion of the result presented. Please, change according to Your results.
Author Response
Rewiew 3
I did enjoy reading this article. The authors bring novel insights in conservation eco-physiology of Primula veris. However, some technical corrections are needed. English improvement is also necessary.
Answer: The authors are thankful for the high evaluation of the manuscript and the remarks given that contributed to its improvement.
The term ex situ collection is used inappropriate, with idea that only outdoor beds are referring to ex situ. In fact, all done and mentioned in manuscript is a mode of ex situ conserving. It needs to be adjusted.
Answer: Of course, all done and described in the manuscript refers to ex situ experiments. However, "ex situ collection" is a widely adopted term used in the literature to distinguish plants grown on experimental filed plot from plants growing in natural populations (in situ).
Introduction is giving very few information of the threats of this species globally and in Bulgaria. It should be added. On the other hand it gives too much space on medicinal properties of target species.
Answer: We edited the Introduction according to your recommendations.
The results presented in rather agricultural way and the aim and focus should be on conservation of the species Primula veris. But it is a matter of the style by the authors.
Answer: The aim of our research is clearly stated at the end of the Introduction. A protocol for ex situ propagation of P. veris has been established, which can be further applied to introduce the species into agriculture. However, the results of the present study are mainly focused on improving seed germination using different approaches. Conservation of P. veris is expected as a result of reduced pressure on its natural populations due to future cultivation of the species.
Conclusion cannot use future tense. The assumption on natural populations of target species recovery in this study cannot be the conclusion of the result presented. Please, change according to Your results.
Answer: We edited the conclusions according to your recommendations.
Round 2
Reviewer 1 Report
Dear authors,
the manuscript has been considerably improved.
Please see attached file for details.
Yours Sincerely,
.

Author Response
Dear Reviewer,
Thank you very much for the careful review of the manuscript. We corrected the technical errors. We removed the Appendix tables as per your suggestion.
Kind regards,
Authors